# Factors Influencing Students' Behavior and Attitude towards Online Education during COVID-19

**Gratiela Dana Boca**

Department of Economics and Physics, Faculty of Sciences, Technical University Cluj-Napoca, 430122 Baia-Mare, Romania; bocagratiela@cunbm.utcluj.ro

**Abstract:** Universities around the world have faced a new pandemic, forcing the closure of campuses that are now conducting educational activities on online platforms. The paper presents a survey about students behavior and attitudes towards online education in the pandemic period from the Technical University of Cluj Napoca, Romania. A group of 300 students participated. The questionnaire was structured in four parts to determine student's individual characteristics, student's needs, students' knowledge in using virtual platforms and students' quality preferences for online education. The students said that online education in a pandemic situation is beneficial for 78% of them. A total of 41.7% percent of students appreciated the teachers' teaching skills and the quality of online courses since the beginning of the pandemic, and 18.7% percent of the students appreciated the additional online materials for study to support their education. However, students found online education stressful, but preferred online assessment for evaluation. This pandemic has led to the new stage of Education 4.0, online education, and the need to harmonize methods of education with the requirements of new generations.

**Keywords:** digital education; management change; student behavior; student attitude

## 1. Introduction

The concept of education has changed dramatically over the last few years, with many questions being raised as to what the best mode of instruction is with the advent of technology and the Internet. The waves of the evolution of education in history begin in the 1780s, with the first wave concerning the individual context of learning and memorization, known as Education 1. The second wave of mass learning appears around the 1900s, known as Education 2. The Internet that allows learning known as Education 3, begin from the 1970s, and has the addition of computers, but only as an interface with students which produces knowledge. Distance learning was first introduced in the 18th century in parallel with the postal service, but it did not pick up steam until communications technology evolved in the 1990s [1]. If we look in time at the stages of the evolution of education, we can see that from a traditional system that focused on books and teaching on the blackboard, over time the use of technology induced a new stage known as Education 4.0, when the computer and the Internet changed the concept of education and the new digital generation offered more possibilities for education.

The future belongs to Education 4.0, as a part of the evolution of education but with a very high impact of digital technology. Empowering education to improve innovation, the transition to the new stage requires the development and harmonization of education systems by employing the new relationship that must be established: student-teacher-technology = smart education and the use of e-education (online, electronic tools). Zhu et al. [2] are supporters of smart education for an environment in which students work as close as possible to reality, which is the reason the education system must combine reality with the virtual world. Zhu et al. [2] and Hartono et al. [3], set out the needs for hybrid education and the term smart learning for students to adapt education to the digital age.

The pandemic contributed to the faster transition to towards the new stage of Education 4.0. Under the imposed conditions, the use of online education was the tool to save and implement digitalization as a beneficial alternative. Education has changed, and online learning is the next big transformation, as Frecker and Bieniarz [1] suggest. One of these advantages is the diversity of educational possibilities and the multiple ways of placing the content. There is also great diversity in terms of assessment, with the teacher having the opportunity to place continuous or summative assessment tests. An assessment by Bond and Lockee [4] and Jackson [5] identifies online needs and considers online professional development courses mandatory in online teaching for higher education faculties. In terms of the impact upon students, it was beneficial to take into consideration their interests and the desire for teachers to post more online courses for the future (Elzainy et al. [6]). Since 2016, a vision for higher education has been designed [7]: a concept put forward by McGee et al. [8], who stress that the training of instructors for online teaching and the preparation of online teaching faculties must also be taken into account. The vision went further, and Rhode and Krishnamurthi [9] developed the concept of self paced training for academic person. During the pandemic, Iwai [10] researched the effects of virtual classroom learning through adaptive learning and virtual reality by use of technology, considering the satisfaction and improvement of skills of the staff and students.

If we take a look in recent time to the evolution concept of online education, we can identify different shapes in the authors' visions, as presented in Table 1.

**Table 1.** Authors different shapes of online education.

| Author | Year | Shape |
| --- | --- | --- |
| Dandara, O. [11] | 2013 | electronic platforms, a means of modernizing educational technologies |
| Allen and Seaman [12] | 2017 | digital learning compass: distance education |
| Espiritu and Budhrani [13] | 2019 | showed the importance of e-learning as a culture |
| Dhawan, S [14] | 2020 | a panacea in the time of COVID-19 crisis |
| Jæger and Blaabæk [15] | 2020 | importance of library because of inequality in learning opportunities during COVID-19 |

Following the European Commission's public consultation [16] in 2020 to develop an action plan for the digitalization of education at EU level, it has been shown that there are still students who have not used distance and online learning tools, but who before the crisis were willing to improve their digital skills.

The idea of improvement and specialization in the field of online education was brought up in the paper by Schmidt et al. [17], and also involvement in blended learning. According to recent studies of Mishra et al. [18] and Bojovic et al. [19], the effects of education during the COVID-19 crisis led to the need for a rapid transition to distance learning. Moskal et al. [20], Moore et al. [21], and Mohr and Shelton [22] established that online education also presents good opportunities for good practices that are necessary for professional development. Based on the action plan of the European Commission for the action of digital education and teaching in higher education [16], the theme of online courses was addressed by Baran and Correia [23] and Baran et al. [24]. At the same time, Espiritu and Budhrani [13] showed the importance of e-learning as a culture. Brinthapt et al. [25] and Elliot et al. [26] considered online learning important for the professional development of staff, following a strategy in [7,16]. The sustainability of online learning is an opportunity, because it is a flexible option that allows the development and improvement of skills.

Along with current digital tools, wider access to the Internet offers different people the opportunity to access higher education. In addition, we can also reduce the carbon footprint by reducing travel and provide a more personalized learning.



## 2. Literature Review

### 2.1. Online Learning and Education

The use of the Internet and state-of-the-art technologies to obtain information for fast communication has become extremely important in the communication and promotion strategy of any university [27]. Communication in the university environment is one of the basic elements on which the student-teacher-university relationship is built. The motivation to approach the communication made by universities starts from the premise that most of the times the students' performances in the learning process and in the integration of the university environment are determined by the way in which the information is made by the universities. Moreover, the COVID-19 pandemic has demonstrated the usefulness of these platforms, as more and more schools move to the red scenario, which means that virtually the entire educational process moves to the online system on educational teaching and learning platforms.

Electronic platforms have a number of advantages over traditional teaching (Elzainy et al. [6]), exploring the impact of e-learning and assessments on students and having observed important changes in improving student's technological skills during the pandemic period. Martin et al. [27] noted the use of traditional assessments to assess the students and course templates, and the processes of quality assurance and surveys, learning analyses and intermediate assessments. Timely response and feedback, availability and regular presence and communication were some of the facilitation strategies used by the award-winning instructors.

The use of educational platforms has allowed finding solutions in the imposed situation and innovating teaching methods and tools in various fields such as geography by Cazacu [28], and medicine by Chatterjee and Chakraborty [29] and Elzainv et al. [6]. Additionally, the use of information and communication with the help of technology has been useful in the medical field, as noted by Grishchenko [30] and Hasan and Bao [31].

The quality of the platform used in the educational process has a favorable effect on the performance of students in online education (Ionescu et al. [32]). According to [33], we can consider that in 2020 the sustainability of online learning offers professionals a flexible option in accordance with their schedule, contact with university staff and platforms for advice and information. Others, such as Singh et al. [34] present the importance of platforms in education but in the same time Diaz and Walsh [35] became advocates for telesimulation based education during COVID-19.

Becker et al. [36] sustain that electronic platforms allow the storage and management of an unlimited number of courses, as well as the storage and management of an unlimited volume of content within a course.

The use of online educational platforms has become a necessity and has spread rapidly since 2020, being the only tools that could be used during the break for online teaching. The influence of smart learning was presented by Budharani et al. [37], and Bojovic et al. [19], like an education in times of crisis: rapid transition to distance learning. However, the use of these educational platforms also has disadvantages, among which we mention: it requires experience in the field of computer use, both by teachers and students, and involves high design and maintenance costs.

The environment in which the students carry out their activity must be as close as possible to reality, which is why it is between the real and the virtual world, and why the educational system must combine reality and the virtual world. Zhu et al. [1]. Martin et and Bolliger [38] agree with the term smart learning to adapt to the digital age Hartono et al. [3], establish the needs for the smart hybrid education.

The COVID-19 crisis has brought to light digital inequalities among students, which is a major risk factor for social vulnerability. Additionally, the inequalities were identified in the research by Beaunoyer et al. [39], because not all students have the same social conditions or lifestyle, and not all have access to the internet or have high-performance digital equipment or have the necessary skills.

Following the study by Jæger and Blaabæk [13], we notice that another inequality is the learning opportunity during and after COVID-19. They highlighted the importance of the library in the learning process, because the backgrounds of students and families are different, as are education and income of the parents.

Based on the action plan proposed by the European Commission for digital education for the period 2021–2027 [16], Zhu and Liu [40], and Hasan and Bao [31] mention that in the pandemic context the innovation in the educational field allowed the identification of the niche elements in digital and post digital education.

The European pandemic COVID-19 has led nationally to the development and taking of rapid and effective measures that have caused significant disruption to education systems, training for both students and teachers but also educators, who at the same time had to adapt to online courses, as Ursan et al. [41] observed.

### 2.2. Online Education and Teachers

Universities and teachers were not completely taken aback by online courses and activities, and Windes and Lesht [42] highlighted the effects of the online courses and their impact on education.

There are currently few studies on the effectiveness of online courses, the teacher-student relationship, and the effectiveness of online assessments. Among those who approached the new topic were Chakraborty et al. [43] and Aguilera-Hermida [44], who noticed that students believe that online education helped them to continue their training and studies during the pandemic with digital platforms, but at the same time to have access to faculty libraries.

Online education for teachers requires time to identify and build the platforms and materials needed, according to Hodges et al. [45]. Bojovic et al. [19] and Chakraborty et al. [43] noted that teachers still lack confidence on online assessment techniques. Aguilera- Hermida [44], argues that teachers' experience can also be closely related with the students' learning experiences. In his opinion, Chakraborty et al. [43] students prefer face-to-face interaction with teachers because teachers do not trust online assessment techniques.

Si et al. [46] stressed the importance of online teaching skills of teachers but also of students who were not prepared for online courses. Teaching platforms have a number of advantages, including real-time access to education but also to the resources placed on these platforms, but also a number of obvious disadvantages, such as the necessary experience and the appropriate means that can involve considerable costs. Lundsford et al. [47], Martin and Bolliger [38] in their research establish the connection between students and professor and the importance of adaptation of methods and strategies in the online learning environment, and also student's involvement [48].

At the same time, the teachers were unprepared for the online activities, and the students also felt unprepared. However, the online platforms allowed at the same time the monitoring of the activities, by visualizing the frequency of the entries and establishing the result of the activities, allowing evaluations on the course but also the efficient final evaluation. The influence of the pandemic after Carroll and Conboy [49], led to new practices that emerged under the pressure of the pandemic ''big bang'' introduction of technology and ''tech driven'' practices. Based on the interviews, Martin et al. [27] found that online instructors create the reverse method through a design taking into account the needs of students.

Polly et al. [50] examined the barriers in the use of digital technologies and the necessary support for academics staff. The barriers identified were the time required to learn new technologies and the time required to learn how to use them in the teaching process. Another factor would be the conflict between the focus on teaching and other service responsibilities, including research. The entire teaching and learning methodology must be transferred online, requiring a systematic reorganization of the learning process through the computer.

The pandemic period provided the opportunity for universities to identify the optimal solutions and to adapt the educational act by opting for the best solution in the given situation. The educational act can be influenced by other factors, such as the communication between students and teachers, identified by Coman et al. [51] or the emotions in the use of digitalization and twitter addiction during the blockade imposed by COVID-19, not only in India, as pointed out by Arora et al. [52].

Perception depending on the field of specialization and the student-teacher relationship were addressed by Trammell and LaForge [53] in their paper, but also the continuous changes necessary for the faculty in the future development of faculties was noted by Stark and Smith [54]. Sheffield et al. [55] instead explored teachers' competencies and attitudes regarding online courses and adapting to students' needs, also Schmidt et al. [17], Trust and Krutka [56] and Rhode et al. [57] identifying what is needed to improve teaching activities and their personalized adaptation.

Other factors were analyzed by Ionescu et al. [32] from three perspectives—teachers, students and parents—which led to the identification of possible psychological effects on students, resulting from the corroboration of social isolation with the online continuation of the educational process.

Electronic platforms have a number of advantages over traditional teaching. One of these advantages is the diversity of educational possibilities and the multiple ways of placing the content. There is also great diversity in terms of assessment, with the teacher having the opportunity to place continuous or summative assessment tests. On assessments used in faculties, Martin et al. [27,38] noted the use of traditional assessments to assess students, course templates and the process of quality assurance and surveys, learning analyzes and intermediate assessments. Timely response and feedback, availability and regular presence and communication were some of the facilitation strategies used by the award-winning instructors. Aguilera-Hermida [44] pointed out that at the same time not only the faculty, but also the students and administrators faced unexpected situations with repercussions in the teaching and learning processes. Some online educational tools facilitate teacher-student collaborative learning as argued by Chakraborty et al. [43] and Adhikary et al. [58]. Important issues are the lack of access to a printer and consumables needed to print worksheets or materials received, children being left alone at home; stress generated by the risk of illness and unemployment.

We can observe that COVID-19 in fact give a push for the acceleration of digitalization in universities all over the world, with researchers trying to identify and diagnose the new provocation, such as Arora et al. [52] in India, Ionescu et al. [32] in Romania, Bojovic et al. [19] in Serbia, Skulmoski and Rey [59] in Germany, as well in the Philippines by Toquero and Talidong [60], in China by Zhang et al. [61] and Islam et al. [62] in Bangadesh.

### 2.3. Students Behavior and Attitude toward Online Education

Since 1986 when the first Technology Acceptance Model (TAM) appeared to identify the factors that affect students behavior and intention to use technology, in time the model was improved and new factors and a more complex investigation was developed, such as in Table 2.

However, today's generations of students were born and started school in the age of the Internet and online browsers, the Google search engine and social media platforms. The digital world is part of the lives of young students, from their first years of life, and online education, through digital applications, is a language of learning that they have been using for a long time.

The means of information used in the university environment are in the process of reconfiguration and development. In the near future, the means of access to information transmitted by universities will be multimedia, mobile and miniaturized.

**Table 2.** Factors influencing student's behavior to use technology.

| Authors | Factors | Results |
|---|---|---|
| Davis [63] 1993 | "Perceived Usefulness" and "Perceived Ease of Use" are the two key factors that affect an individual's intention to use a technology | to investigate the impact of technology on user behavior |
| Liu et al. [64] 2010 | namely Online Course Design, User-interface Design, Previous Online Learning Experience, and Perceived Interaction | to investigate the impact of technology on user behavior |
| Al Kurdi et al. [65] 2020 | involved E-learning Computer Self-Efficacy, Social Influence, Enjoyment, System Interactivity, Computer Anxiety, Technical support, Perceived Usefulness, Perceived Ease of Use, Attitude, and Behavioral Intention to Use e-learning | a suitable theoretical tool to comprehend the acceptance of e-learning by users |
| Al Kurdi et al. [66] 2020 | "social influence, perceived enjoyment, self-efficacy perceived usefulness, and perceived ease of use" are the strongest and most important predictors in the a virtual E-learning atmosphere intention of and students towards E-learning systems | to improve ongoing interests and activities of university students in a virtual E-learning atmosphere |
| Mailizar et al. [67] 2020 | The model consists of six constructs: system quality, e-learning experience, perceived ease of use, perceived usefulness, attitude toward use, and behavioral intention. | to improve the understanding of students' intention to adopt e-learning. |

More and more universities are adopting coherent strategies for integrating technology into the educational process and the media used in both internal and external communication tend to migrate to online. Student's opinion regarding the digital learning and the impact in their daily life was investigated by Martin et al. [27], Bao [68] and Chakraborty et al. [43], but also, they take in consideration the student's opinion about online education in the pandemic period.

The transition to online education and students' intention to use online education was a challenge during the pandemic, and the studies provided information that will underpin future strategies for developing education and improving the quality of online education use and involvement of both actors involved in the system—students and teachers.

There is a wide range of factors that have been taken into account in order to identify student behavior and attitudes, as shown in Table 3.

**Table 3.** Online education factors influencing the student's behavior and attitude.

| Authors | Year | Factors |
|---|---|---|
| Lee, J.W. [69] | 2010 | online support service quality, online learning acceptance, and student satisfaction |
| Hung and Jeng [70] | 2013 | age, online teaching experience, implications of the findings were discussed the lecturer's competence, the lecturer's attitude towards learning, and the nature of the subject knowledge's attitude and practices, |
| Hatabu et al. [71] | 2020 | the frequency and activities of information acquisition, the correct explanation of the information and willingness to collect anxiety information |
| Adil Zia [72] | 2020 | attitude, curriculum, motivation and technology training |
| Alzahrani et al. [73] | 2021 | service quality, information quality and self-efficacy, satisfaction |
| Yunus et al. [74] | 2021 | the effort expectation, the performance expectation, social influence and facilitating conditions in using the online education |

Studies on e-learning and the impact on students were conducted recently in 2020 by Bao [68], Islam et al. [62], Essadek and Rabeyron [75] and Paea et al. [76] showing that the new impact of online education among students has fostered depression and anxiety in the pandemic period. Cao et al. [77], suggested that the mental health of college students should be monitored during epidemics because of the pressure and stress. Mishra et al. [18] found that due to the limiting of travel, i.e., the academic exchange programs of students

and staff between universities, there was also a deterioration of academic research actions and activities in education.

Students' behavior and attitudes toward online education and the use of digital platforms during the pandemic have led many researchers to conduct new studies to identify the new environment and these factors.

### 3. Materials and Methods

The pandemic did not find the university and teachers totally unprepared, because a platform (edu.utcluj.ro) was created long before for the students from the low frequency system, where the courses, seminar materials and other information necessary for those who work and attend the faculty were posted. The platform was also for full-time students to be able to send homework proposed by teachers or for teacher-student information.

During the pandemic, the platform was improved, and another one, the Knowledge Base (kb.cunbm.utcluj.ro) platform was created, allowing the students from the economic specialization to access them and to build the teacher-student bridge.

In addition to the digital platform for seminars, homework, additional materials, systematic or periodic evaluation, some teachers have chosen other teaching tools, such as ZOOM or Microsoft Team, for a more attentive and beneficial communication. In this way digital technology has brought a plus to the educational act.

The transition from the traditional classroom teaching system to online education, keeping to the schedule, was made gradually at the beginning, and the courses were held online and with face-to-face seminars, but with the lockdown during the pandemic everything moved to the total online system. The academic staff put in practice Hrastinski [78] and Flora Amiti [79] suggestions, by keeping the teaching systems but using actual modern digital tools, all three modes of online learning: asynchronous, synchronous and hybrid.

The questionnaire was applied between October–November 2020 when the pandemic restriction allowed students' to participate in specific situations in activities at the university. The sample of 300 respondents consisted of graduate students in the final grades and master students from the department of economics, because they could compare the two methods of education before and during the pandemic. The items were established to determine the factors which are influencing the students behavior and attitudes concerning online education.

The study wanted to identify students' behavior and their attitudes in the new context of online education. The structure of the questionnaire and factors influencing their behavior are presented in Table 4.

In order to obtain the necessary data, the target group was selected on a voluntary basis, and the period was targeted respectively when students chose their place, the period of the internship or the coordinators of the license or dissertation.

All students voluntarily consented and confirmed their participation in the study after being previously informed of the purpose of the study.

The study allowed the analysis of the behavior and association with student's attitude towards online education.

The first part of questionnaire tried to identify the socio-demographic characteristics of the respondents, three questions were used regarding gender, age and education level. The education level includes two categories: bachelor and master students. The age groups were established between 18–26 years old, 26–32 age, 32–38 age and 38–42 age.

The second part of the questionnaires were focused on students behavior related to their needs: time spend using online education tools. Additionally, the questionnaire establishes and identifies the frequency of hours of consumption of virtual tools used in education, which was measured by asking respondents: "How often do you enter weekly online?". For the response categories for their time spent online, read Supplementary Materials (1 h, 10 h, more than 10 h) and the emphasis they put on the quality of courses and how this affects students behavior.

**Table 4.** Questionnaire structure.

|  | Questions | Items | Factor |
|---|---|---|---|
| 1 | Age | I1 | Individual Characteristic |
| 2 | Gender | I2 | |
| 3 | Education level | I3 | |
| 4 | How many hours are you spending weekly online? | H | Needs |
| 5 | Do you find digital learning and platforms useful? | U | |
| 6 | How many hours do you devote to individual study? | S | |
| 14 | How often you enter online | F | |
| 8 | What kind of examination do you prefer (online) | E1 | Knowledge |
| 9 | What kind of examination do you prefer (writing) | E2 | |
| 10 | What kind of examination do you prefer (test) | E3 | |
| 11 | What kind of examination do you prefer synthesis | E4 | |
| 12 | What kind of examination do you prefer portfolio | E5 | |
| 7 | Do you read the specialty materials? | R | Quality |
| 13 | How you appreciate the online courses quality | Q | |
| 15 | How do you consider the learning activity | QE1 | |

In the third section of the questionnaire, students' knowledge was established, to determine students culture on virtual media, their abilities to use the modern tools, the benefits of education online, how important for students their education is, and their orientation between the traditional and new types of education. The answer categories for the open question about type of examination preferred by students offered the respondents the possibility of stating their favorite one. The key element was the item regarding the final evaluation of their activity, respectively the way of conducting the final evaluation exam. They were able to choose from several types of assessment so as to see which type of assessment is considered the most optimal assessment of their knowledge. Regarding the students preferences type of examination, it was possible to choose between online, writing, multiple choice test, syntheses and portfolio. A Likert scale was used for the study, starting from a score of 1 representing the student disagreement—"Not at all'' and up to a score of 5—"Very well", representing the student's strong agreement.

The last part of the questionnaire included three questions that identify students satisfaction about the quality of online courses, the quality of learning activity on the pandemic period and the quality of specialty materials. From research the article wanted to obtain and know students interaction with the professor through an online platform quality with the item "*How you appreciate the online courses quality*", how useful online education iswas identified with the item "*Do you find digital learning and teaching platforms useful*?"

### 3.1. Sample and Measurement Tool

To determine the factors that influence students' behavior and attitudes during the pandemic due to the use of online education, a statistical analysis was performed using SPSS software applied to the given database, as well as the independent simple t tests and hypothesis testing. The proposed model and the correlations between the items of the questionnaire were made using the solutions of the Lisrel 8.7 program.

### 3.2. Purpose of the Study

The study presents the following four factors that influence students behavior that were taken into consideration:

1.  Student's individual characteristics (age, gender, education level);

2. Student's needs: frequency of entering the online platform (F), hours spent on the virtual platform (H), hours to study and learn from materials proposed by teachers (S), and how useful are digital platforms for their needs and for their better understanding (U);

3. Student's knowledge regarding the type of knowledge, evaluation portfolio, syntheses, test, written examination and online examination (E1, E2, E3, E4 and E5);

4. Student's perception about quality of online education: courses quality (Q), education and learning quality (QE1) and quality of materials presented and information (R).

The research model from Figure 1 is based on the research objectives and hypothesis.

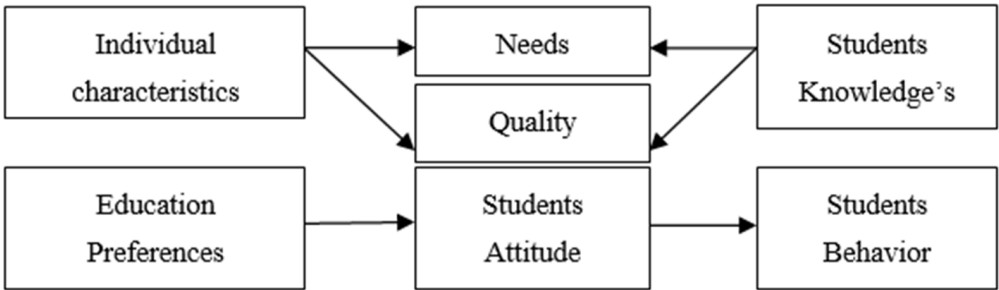

**Figure 1.** Research model to examine the students behavior and attitude. Source: by author.

The hypotheses tested on the attitude of students in the present study are:

**Hypothesis 1 (H1).** *Students preferences have a significant effect on their attitudes towards their online education.*

**Hypothesis 2 (H2).** *Students preferences have a significant effect on their attitudes towards needs.*

**Hypothesis 3 (H3).** *Students preferences have a significant effect on their attitudes towards knowledge.*

**Hypothesis 4 (H4).** *Attitudes of students concerning their needs of evaluation has a significant effect on their behavior.*

**Hypothesis 5 (H5).** *Attitudes of students towards online education have a significant effect on their behavior.*

Taking into consideration Saraçli et al. [80] and Gumus et al. [81], the Cronbach's alpha were calculated as a quality instrument to analyze the items. The total Cronbach's alpha value of scale was calculated as 0.72 which is statistically one of the indicators that shows that the reliability of the scale.

With $\alpha$ value in interval $0.8 \leq \alpha \geq 0.7$ interval following the statistical rule [82,83] we can consider acceptable and good enough the model. After reliability analysis, exploratory factor analysis (EFA) was applied over the four factors: Individual characteristics of students, Needs, Knowledge and Quality.

The root mean square error (RMEA) of approximation value was 0.63: normed fit index (NFI) was 0.93, non-normed fit index (NNFI) was 0.96, comparative fit index (CFI) was 0.90, goodness of fit index (GFI) was 0.90 and adjusted goodness of fit index (AGFI) was equal to 0.88. If we compare the goodness of fit, the model is in acceptable fitness.

Using the Schermelleh-Engel and Moosbrugger structural model [84], the next step was to compare with the program solution, and for the structural equation model we obtained the values shown in Table 5.

**Table 5.** Structural equation model fit.

| Criteria | Acceptable Fitness | Model |
|----------|--------------------|-------|
| RMSEA | $0.05 \leq RMSEA \leq 0.10$ | 0.63 |
| NFI | $0.90 \leq NFI \leq 0.95$ | 0.93 |
| NNFI | $0.95 \leq NNFI \leq 0.97$ | 0.96 |
| CFI | $0.95 \leq CFI \leq 0.97$ | 0.90 |
| GFI | $0.90 \leq GFI \leq 0.95$ | 0.90 |
| AGFI | $0.85 \leq AGFI \leq 0.90$ | 0.88 |

Source: adaptation after Schermelleh-Engel and Moosbrugger [84].

## 4. Results

Based on the data obtained by applying the questionnaires following the data analysis, we can draw the following results regarding the factors that influence the behavior and attitude of students towards online courses. For the first part of questionnaire regarding the students characteristic in an equal percentage there are from master's and bachelor's degree. The target group was made up of 56.66% females and 43.34% males, aged between (18–26) (66.33%), ages (26–32) (17%), ages (32–38) (11.34%) and ages (38–42) (5.33%) (Table 6).

**Table 6.** Distribution according to respondents education by age and gender.

| | | Age | | | | Cumulative Percent |
|---|---|---|---|---|---|---|
| | | **18–26** | **26–32** | **32–38** | **38–42** | |
| Gender | female | 29 | 13.33 | 9 | 5.33 | 56.66 |
| | male | 37.33 | 3.67 | 2.34 | 0 | 43.34 |
| Total | | 66.33 | 17 | 11.34 | 5.33 | 100 |
| Education license master | | 22.33 | 13.67 | 8.67 | 5.33 | 50 |
| | | 44 | 3.33 | 2.67 | 0 | 50 |
| Total | | 66.33 | 17 | 11.34 | 5.33 | 100 |

Regarding students attitude about online education, 43.3% of respondents consider that being face-to-face and physical presence in amphitheaters is beneficial, and only 33% of students considered that online education is better. A total of 23% of respondents considered that online education can be an annexes or complementary education together with face-to-face lectures to improve education.

At the same time 41.7% of students believe that the pandemic situation has led to improved teachers' skills through new teaching methods and techniques adapted to the online environment since the beginning of the pandemic, and 34.7% of respondents consider satisfactory the new tools used by professors in their online lectures. However, 23.7% of students consider online courses, seminars and evaluation as the only beneficial solution in the pandemic situation created.

Regarding the items about the reading materials from platforms and online education tools used by professors, such as slideshows, notes, problem solving, student's perception is positive at 63.3%, feeling that the volume of information is adequate and is available online.

Another barometer that come to support the students' behavior is regarding the attention paid to the additional materials, we found that professors use different platforms such as ZOOM, Knowledge Base, together with classical tools such as Power-Point and Prezi to follow, which make sessions more interactive, as appreciated by the students. At the same time, some courses of specialization require supporting quality interaction (Singh et al. [34]).

Table 7 presents that 58.66% of respondents spend weekly between 1–10 h on the platforms, and 24.7% more than 20 h.

**Table 7.** Time spend by students online.

|  | Female | Male | Cumulative Percent |
|---|---|---|---|
| How many hours are you spending weekly online |  |  |  |
| Non | 14 | 36 | 16.67 |
| 1–10 h | 98 | 78 | 58.66 |
| 10–20 h | 58 | 16 | 24.67 |
| Total | 170 | 130 | 100 |

As a result, 49.3% percent of students read additional materials proposed by teachers if the information provided through online courses is sufficient, which shows that students are aware of the situation and the new context. A total of 16.7% percent of students do not actually participate in online courses when they are scheduled, due to the fact that they either work or cannot access the Internet with connection problems, but have access to the information on the platforms appreciating this in online courses. A total of 56.66% of female respondents are more active on platforms in comparison with males at 43.33%, maybe because they are more curious and conscientious.

*Factors Influencing Students Behavior*

In Table 8, we can see the correlation between respondent's age and their needs and preferences regarding the type of evaluation.

**Table 8.** Correlation between respondents age and examination needs.

|  | Age | | | | Cumulative Percent |
|---|---|---|---|---|---|
|  | 18–26 | 26–32 | 32–38 | 38–42 |  |
| What kind of examination do you prefer |  |  |  |  |  |
| Portfolio | 19 | 5 | 2 | 0 | 8.67 |
| Synthesis | 17 | 4 | 3 | 1 | 8.33 |
| Test (multiple choice) | 71 | 20 | 15 | 9 | 38.33 |
| Written exam | 17 | 1 | 2 | 3 | 7.67 |
| On-line exam | 65 | 20 | 10 | 2 | 32.33 |
| Total | 189 | 50 | 32 | 15 | 95.33 |

The students prefer the grid test in a percentage of 38.33%, online assessment in a percentage of 32.33%, and portfolios in a percentage of 8.67%. We can conclude that the student's behavior was influenced by the pandemic isolation and also the communication, they prefer the short answer, i.e., fast communication which does not involve too much of their participation.

The results present a young generation accustomed to the Internet and who prefer online examination for 28.3% and verification in the form of tests for 30.33%. For a generation known as native to the digital life, which grow up with new technology the Internet, it is part of their everyday life, so the impact of online education was something normal. For the mature generation, the new technology is a problem because they do not have the necessary skills to use the latest generation technology.

From the gender point of view, the results from Table 9 show a highest percent of feminine preferences for online examination at 17% and a preference for multiple choice tests evaluation at 23.33%.

For male respondents, the preferences are 15.33% for online and multiple choice tests and 4% for synthesis and written exam for evaluation. We can conclude that students attitude towards digital platforms and online education influence their behavior in the function of individual characteristics of age, gender and their abilities and skills with the new technology generation.

**Table 9.** Correlation between respondent's gender and kind of examination.

| | Gender | | Cumulative Percent |
|---|---|---|---|
| | Female | Male | |
| What kind of examination do you prefer | | | |
| On-line exam | 51 | 46 | 32.33 |
| Written exam | 11 | 12 | 7.67 |
| Test (multiple choice) | 70 | 45 | 38.33 |
| Synthesis | 13 | 12 | 8.33 |
| Portfolio | 16 | 10 | 8.67 |
| Total | 161 | 125 | 95.33 |

## 5. Discussion

### 5.1. Variables Correlation

To determine the correlation between the variables and to obtain the degree to which online education influences students' behavior, the classification and regression tree (CRT) analysis was used.

Figure 2 shows the classification of students' behavior regarding the information they obtain using supplementary materials for a better understanding of the information received but also the benefits of online education.

Even if the courses and seminars took place online, the teaching staff followed the same steps of the traditional system, with the recommendation of some additional materials for the students.

A total of 18.7% of students apply these additional materials for a good understanding or for information. For a percentage, the technology was a form of rescue in the pandemic situation, for 18% of students this was out of curiosity, and 63.3% do not apply because they consider that they receive enough information based on the assessment and materials on the platform provided by the teacher.

The online courses are considered beneficial by 41.3% of the students, appreciated especially by the working segment, as they have part time jobs, due to the flexibility of the program and the possibility to access the information when they have time.

On the other hand, 64.7% of those who are pursuing a bachelor's degree appreciate this type of course, but for those at the master's level, 39.3% find it as beneficial.

The data of this exploratory study highlights the fact that students go through a process of adaptation and learning, constituting an indirect but favorable argument.

Figure 3 represents the classification for students' behavior using online education. It is a weekly or daily routine for 24.7% of respondents to spend more than 10 h on the platform, and 58.7% of students are spending between 1–10 h daily.

Additionally, 26% of students are male and 32.7 % are female. Taking into account the time spent by students online on educational platforms, we notice that a percentage of 57.8% of students spend between 11–0 h per week, of which 43.2% are at the bachelor's level and 56.8% are at the master's level, we note that they use the information on the platforms put on display by the teachers.

The highest value was obtained for undergraduate students who spend more than 10 h on the university platform, respectively for the field of management and economics, at 24.4%.

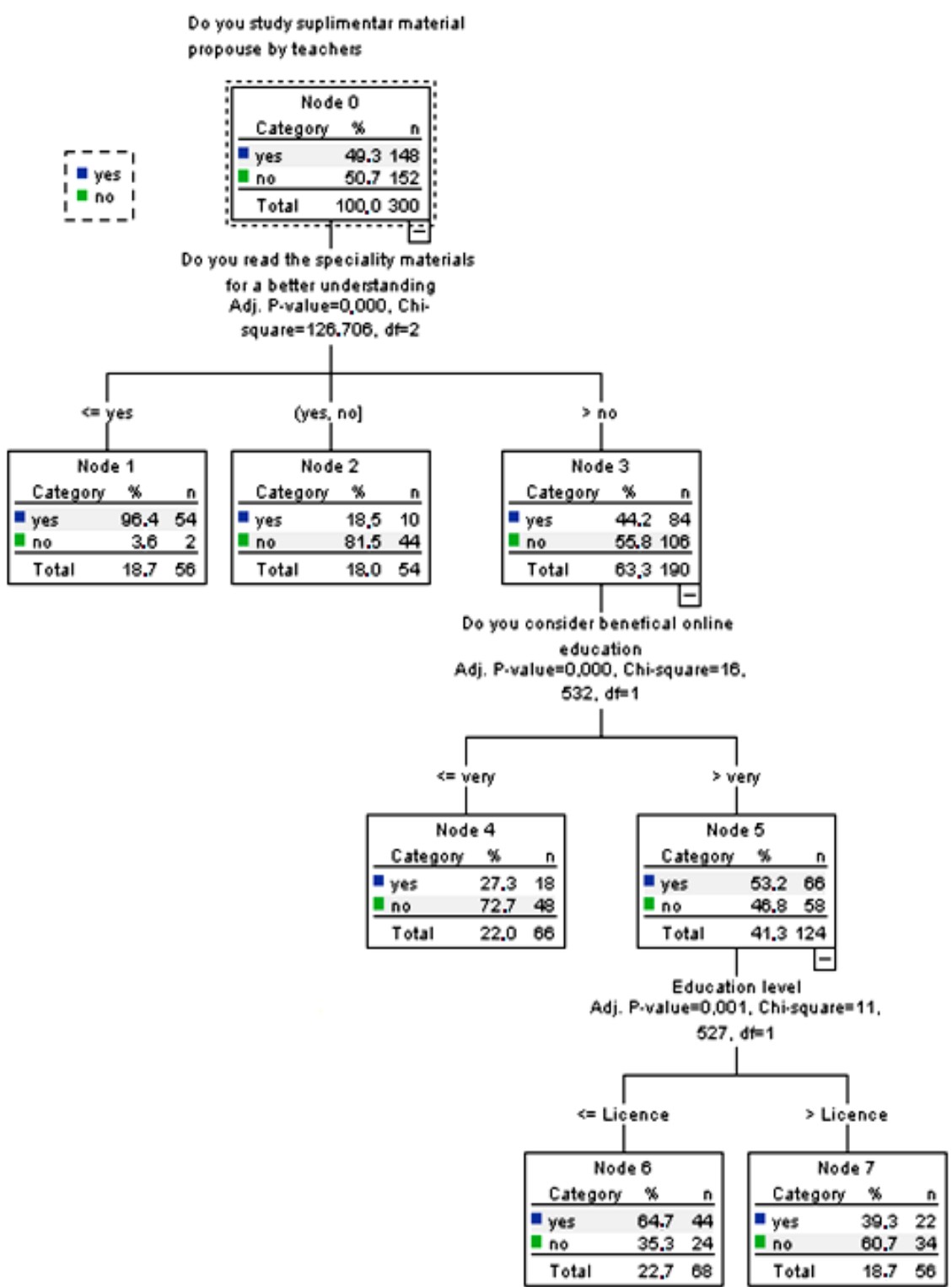

**Figure 2.** Classification for student's behavior using supplementary materials. Source: By author.

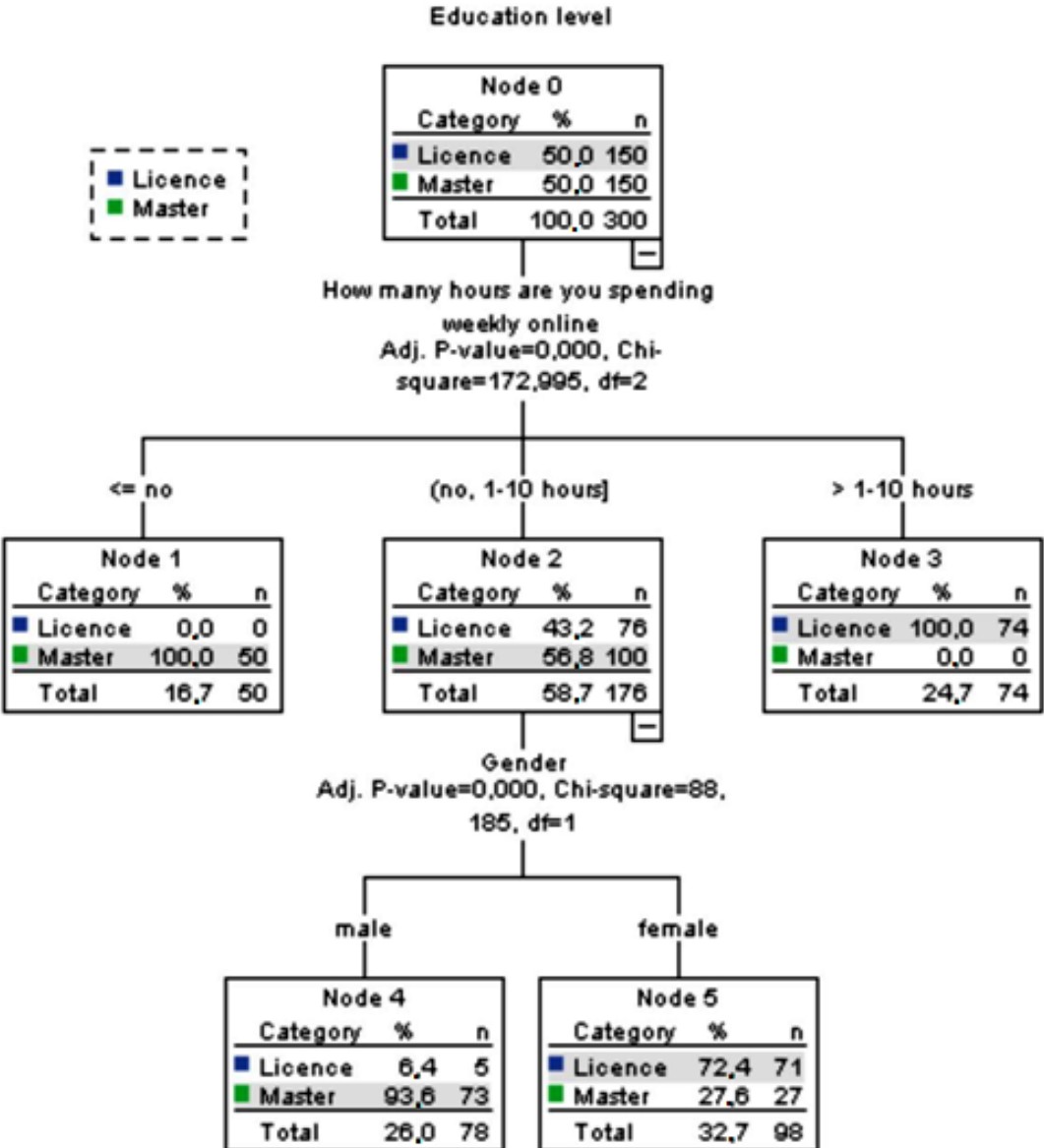

**Figure 3.** Classification for student's behavior using online education.

*5.2. Correlations between Items*

Using the database obtained from the 15 items of questionnaries from SPSS using the Lisrel 8.7 program, four groups of students with similar profiles regarding the online education were identified.

The model take into consideration the students characteristics (gender, age, education level), how the students are involved in diferent activities for a better asimilation of information and their knowledge from different activities in their culture (U, S, H, F), student needs for evaluation (E1, E2, E3, E4 and E5) and student quality satisfaction about online education (R, Q, QE1). The program solution for the students behavior model is presented in Figure 4.

Students behavior using online education during the pandemic period gives us the following correlations.

The highest positive value of 0.89 was obtained for the correlation between Individual characteristics and students Knowledge. The results represent students from the Baia Mare faculty, the future managers and economists or entrepreneurs, having strong knowledge of the online education and of how they are evaluated.

Another strong relation, of 0.07 value, was between students Needs and Quality of education, in our case online education, so the student involvement and virtual approach is necessary and beneficial by accessing the materials provided by teachers on platforms so that the educational process does not lose quality.

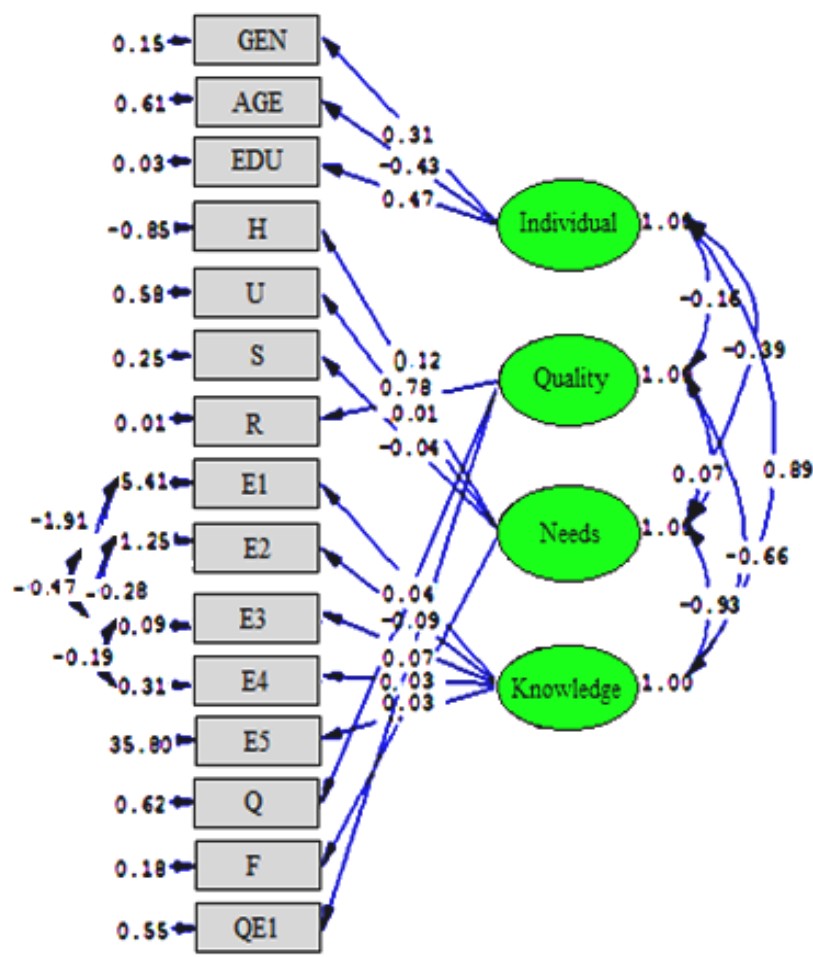

**Figure 4.** Factors influencing students' behavior using online education.

By using the online platform, from the Quality characteristic point of view, the reading (R), of supplementary and speciality materials obtained the strongest value of 0.78.

Additionally for Individual charactersitics of students the level of education EDU obtained the bigest value of 0.47,that means the students are informed and enjoy use of the platforms and being involved in online education, with the finalization on specialization.

For Needs, the highest value of 0.12 was obtained by the number of hours spent by students online (H), followed by the frequency (F) with which they access the platform, with a value of 0.03.

For Knowledge, the highest value of 0.07 was obtained by evauation using mulitple choice test and the lowest by written examination, with a value of −0.9.

In conclusion, between students individual characteristics and students knowledge's using digital platforms, there is a strongest correlation and the quality of educational process, it is not influenced by the individual characteristics. Student's attitudes towards online education is influenced by their needs and the platform quality improves student's knowledge and behavior.

### 5.3. Conventional Students Cluster

In order to be able to study students' behavior regarding the online education, a cluster analysis was also performed, this time taking into account the order of their preferences. The sample of 300 students was subjected to a k mean and hierarchical grouping, we proceeded to identify four groups of students with similar educational profiles. According to the features of the adult students (Table 10) four clusters were identified: Needs of students (characterized by online benefits, and individual study of specific materials), students Knowledge (characterized by the hours spent weekly online) Quality for students (characterized by quality of materials and speciality materials,), students Preference for evaluation (characterized by different types of online evaluation, multiple choice, portfolio and written version).

**Table 10.** Students conventional clusters.

| Items | Clusters | Number of Students |
|---|---|---|
| Gender | | |
| Education level | | |
| Do you consider beneficial online education | Needs | 73 |
| Do you study supplementary material propose by teachers | | |
| Age | Knowledge | 73 |
| How many hours are you spending weekly online | | |
| Do you read the specialty materials for a better understanding | Quality | 32 |
| What kind of examination do you prefer | Preferences | 122 |

Cluster 1: for 73 students Needs of online education and supplementary materials posted on the platform by the teachers are important, expressing their attitude to spend time and hours to collect the information.

Cluster 2: for 73 respondents Knowledge is important, the number of hours spent online or on platforms depended on the information and their connection with teachers and participate in different activities. Age confirms once again student's responsibilities for their professional preparation and skills impact in using new technology.

Cluster 3: Quality obtained the lowest value for using additional materials. The low value shows that teachers provide enough information through online courses and materials posted on platforms. The university's own platform and for each specialization allows students access at any time to have access to information.

Cluster 4: the highest value, for 122 students, was obtained in terms of the cluster for the student Preferences for the final evaluation, during examination or periodical evaluation. Among the verification options proposed: online, multiple choice, test and written verification, students have shown that assessment is very important for them in the time of pandemic also.

## 6. Conclusions

The article presents the results obtained following the application of questionnaires applied to identify student's behavior and attitudes towards online education during the COVID-19 pandemic.

Based on the literature, the results were able to create a student profile model and establish the factors which influence student's behavior and attitudes concerning online education. Online education has been a great challenge for both teachers and students. At present, education is still in a period of adaptation, of identifying the factors that influence the educational act for an as yet unexplained period. The COVID-19 pandemic brought for

the first time the widespread adoption of online education around the world, making it a necessity in difficult times.

The four factors taken into consideration for the model were the individual characteristics specific to each student, the students' knowledge, the students' needs and the preferences for the quality of online education influencing the students' behavior and attitude.

The student's behavior is influenced by their attitude regarding their needs and quality digital education. The students' preferences for quality platforms and materials in the changes of this period confirm the hypothesis and the model.

Students' have represented that teachers are those who adapt and reformulate their habits, making them closer to students through the digital environments of the future, even if there is further physical distance, which contributes to a categorical evolution of university education.

The feedback of the questionnaire confirms that there is a strong connection between student's needs and the quality of education and the teaching process that influences the student's behavior, and not only in the pandemic period [3,4,11].

The results confirm that the students' obtained knowledge by using online education during this pandemic, which is a useful lesson for future demands, confirming other researcher's results [43,48,77].

The study identified students' needs, and this data suggests that it is not very realistic to start from the assumption that switching to teaching exclusively online can be done easily. The study identifies and confirms also that there are some inequalities regarding Internet access (no telephone signal, or do not have a computer/laptop/tablet/mobile phone, as well as a fairly low level of digital skills) [8,18]. Students have begun to notice that it is possible to put in more effort and have more time to attend courses through online digital tools, even if they are in isolation, at home.

Additionally, the individual characteristics presented similar preferences regarding the digital education and needs, as the correlation between individual characteristics and needs was confirmed [37,38,40].

The data obtained on the basis of statistical observations can highlight the attitude towards the use of a mixture of educational tools to ensure a new orientation towards a new vision for the future of Education 4.0, that changes their behavior towards online education not only in crises or pandemics (Table 11).

**Table 11.** The transfer to Education 4.0.

| PAST | FUTURE |
| --- | --- |
| Traditional | Digital |
| Schedule | Flexible program |
| Staying in university | Staying at home |
| Input focus | Output focus |
| Face to Face | Communication tools |
| Focus on knowledge | Adaptive learning |
| Present information | Share information |
| Classroom | Virtual Classroom |
| Medium condition | Comfort environment |
| Rigidity | Flexibility |
| Presentation centered | Student centered |
| Socialization | Isolation |

The study also confirmed that students prefer online education, but cannot replace the classic face-to-face method. However, students believe that there are opportunities for improvement and easing of the educational process.

The study can be a provocation for teachers to adapt teaching methods and techniques for online education using the new digital generation to use new technical methods such

as inverted class, case studies and playing games which can be used for online technology, such as virtual platforms [39,41,42].

*A SWOT Analyze of Online Education*

As a final conclusion, a SWOT analyses has been made, with the following conclusion:

The study captured, as a last dimension, the students' opinions regarding the present educational context, a unique context for both teachers and students.

Thus, the mood that the students declare is a good one, they are connected to the information about the current situation, and they consider that the return to previous interactions will occur relatively quickly, but at the same time, they declare a low involvement in community volunteering activities.

It is natural, in conditions of uncertainty, for the decision to involve the community of whether to carry on the alternative, limiting the risk.

A strong element reported by students was that the use of digital resources in learning is perceived as a positive thing; also, media tools are preferred by the Facebook generation of students.

A signal is the new profile of the outlined student in the new educational context and the students' perception consists in the following weak elements:

- The topics and tasks proposed to the students following the courses and seminars are several;
- Teachers tend to monitor the student's progress;
- There is final and ongoing evaluation for the continuity of the learning process;
- The transfer of educational activities in the online environment rather negatively affects only the seminar activities;
- The teacher and student no longer interact enough;
- Total transfer to virtualization of activities.

The most important perceived disadvantages of transferring educational activities in the online environment are related to the long time spent in front of the computer to participate in teaching activities and to solve the tasks received, and the perceived excessive volume of homework and tasks. The s perceived as non-problematic can be listed as: the ease of use of digital tools and platforms, but also the accessibility of teaching and learning resources provided by teachers.

An opportunity for generations of students is that the Internet and online browsers, the Google search engine and social media platforms are the environment in which their generation was born and raised. The digital world is part of the lives of young students, and online education, through digital applications, is a language of learning that they have been using for a long time.

We can also mention that students have started to notice that it is possible to put more effort and have a longer time to participate in courses and applications through online digital tools, even if they are in isolation, at home.

This study was conducted in the Technical University Cluj Napoca, Romania, and the model and results can be used by other universities to identify students behavior and attitudes toward online learning and methods which can be used in future activities or other pandemic situations.

**Funding:** This research received no external funding.

**Institutional Review Board Statement:** Not applicable.

**Informed Consent Statement:** Not applicable.

**Data Availability Statement:** Data sharing not applicable. The data are not publicly available due to participants' privacy.

**Conflicts of Interest:** The author declares no conflict of interest.

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
