# Peer review of "Factors Influencing Students’ Behavior and Attitude towards Online Education during COVID-19"

_sustainability, doi:10.3390/su13137469_

Round 1

Reviewer 1 Report

The theoretical framework is actual and interesting. New educational practices are emerging under the pressure of the pandemic, and it is necessary to analyse the experiences of the students to improve them.
It is clear that the COVID-19 has accelerated the digitization of the teaching-learning process, so this study is well justified within this context.

However, the article has 3 weak points in relation to its design:

1) The objective of the study is not clear enough.

2) The structure of the questionnaire used is not understood and the variables evaluated are not clear.

3) The relationship between the data collected and the conclusions can be improved. The wording of the conclusions is confusing. There is not a section for the conclusions and discussion.

It is needed to clarify the objective of the study. As per the article, the purpose of the study is to know the opinions of the students about digital learning in their daily life (217) and/or their opinion about digital learning during the pandemic period (228). However, in the materials and method section, it is only specified that it is intended to know the student's appreciation during the pandemic period while the course was suspended (229), and it is not specified the digital learning in their daily life. Then, these are two different objectives that must be clarified.

It must be determined the teaching-learning context for better understanding how the students were affected by the COVID pandemic situation since, as the article indicates, it was under this situation when the study was carried out. For this, answers to the following questions would be needed but, however, they are not clarified in the article: How were online classes handled during the pandemic period? Were they synchronous or asynchronous? Was any online platform used at the university? Was the evaluation of the subjects in the university/faculty modified?

In the article, it is mentioned that to achieve this objective two processes have been carried out: interviews and a questionnaire.

In relation to the interviews carried out during the period when the face-to-face classes were on hold: What was the purpose? What questions were asked? Was there a record of the most important data carried out? How many subjects were interviewed? What was the way to qualitatively analyze the data? All of these items should be mentioned in the article to better understand the value of these interviews.

In relation to the questionnaire, it would be important to mention in the article the guidelines given to the students about data entry and if they were informed that this questionnaire was in the context of a research. Also, the questions are all written in the present, not in the past, then if the questionnaire was completed by the students after the classes were on hold (October-November 2020), how the author can guarantee that their responses are in relation to the pandemic period and not to their current daily life.

The structure of the questionnaire is not properly explained and the variables evaluated are confusing. For example:

  • In Table 2, which shows the structure of the questionnaire, Students Knowledge items are discussed in the second part. However, in the explanation of the table content, part 2 refers to Student's Needs (243), which is in fact the part three in the table, and even later (270) it is referring to Student Behaviour, which is part four in the table.
  • In the explanation (page 7, 288), it is explained that part 4 refers to the preferences in the evaluation, while in the table it refers to the Students Perception.
  • In the development of the explanation (270) it is specified that the attitudes of the students are established (it is not named in table 2) what questions exactly evaluate the attitude?
  • Following the explanation of the questionnaire (288), the author comments again that there are three items in relation to the evaluation, but in the structure of Table 2 the three final questions do not refer to the evaluation and are related to the student's perception in relation to the courses.

In conclusion, when reading the design, there are doubts in relation to the structure of the questionnaire and whether the questions asked really evaluate what it intends to measure. It is recommended to clearly define each variable. Validation by an expert would have been necessary to assess the questionnaire and to make sure that the items evaluate what is really intended in relation to online education.

Also, it is necessary to explain the differences between each variable and with what specific items each is evaluated.

-Reference number 11 is cited with different authors in the text (79) (213) see (669).

Author Response

Response to Reviewer 1 Comments

Dear Editor

Dear Reviewer

I very much appreciated the encouraging, critical and constructive comments on this manuscript by the reviewer. The comments have been very thorough and useful in improving the manuscript. I strongly believe that the comments and suggestions have increased the scientific value of revised manuscript by many folds. I have taken them fully into account in revision. I am submitting the corrected manuscript with the suggestion incorporated the manuscript. The manuscript has been revised as per the comments given by the reviewer, and our responses to all the comments are as follows:

The theoretical framework is actual and interesting. New educational practices are emerging under the pressure of the pandemic, and it is necessary to analyse the experiences of the students to improve them.It is clear that the COVID-19 has accelerated the digitization of the teaching-learning process, so this study is well justified within this context.

However, the article has 3 weak points in relation to its design:

1) The objective of the study is not clear enough.

2) The structure of the questionnaire used is not understood and the variables evaluated are not clear.

3) The relationship between the data collected and the conclusions can be improved.

Point 1. The wording of the conclusions is confusing. There is not a section for the conclusions and discussion.

Response 1:

I create separate Section   5. Dissscusion and  6. Conclusion

  1. Discussion

5.1. Variables Correlation

To determine the correlation between the variables and to obtain the degree to which online education influences students’ behavior, the classification and regression tree (CRT) analysis was used. Figure 2 shows the classification of students’ behavior regarding the information they obtain using supplementary materials for a better understanding of the information received but also the benefits of online education.

Figure 2. Classification for students behavior using suplimentary materials. Source: By author.

Figure 3, present the classification for students’ behavior using education online. Weekly or daily is a routine for 24.7 % of respondents to spend more than 10 hours on platform and 58.7% of students are spending between 1-10 hours daily.

Figure 3. Classification for student’s behavior using online education.

Additionally, 26% of students are male and 32.7 % are female. Taking into account the time spent by students online on educational platforms, we notice that a percentage of 57.8% of students spend between 1-10 hours per week, of which 43.2% are at the bachelor's level and 56.8% are at the master's level, we note that they use the information on the platforms put on display by the teachers.

The highest value was obtained for undergraduate students who spend more than 10 hours on the university platform, respectively for the field of management and economics in a percentage of 24.4%.

5.2. Correlations between Items

Using the database obtain from the 15 items of questionnaries from SPSS using the Lisrel 8.7 program, four groups of students with similar profiles regarding the online education were identify.

The model take into consideration the students characteristics (gender, age, education level), how the students are involved in diferent activities for a better asimilation of informations and their knowledges from different activities for their culture (U, S, H, F), students needs for evaluation (E1,E2,E3,E4 and E5) and students quality satisfaction about online education (R, Q, QE1). The program solution for the students behavior model it is presented in Figure 4.

Figure 4. Factors influencing students’ behavior using online education

           Students behavior using online education towards pandemic period gives us the following correlation:

The highest positive value of 0.89 was obtained for the correlation between individual characteristics and students knowledge. The results present that students from Baia Mare faculty, the future managers and economists or entrepreneurs, have strong knowledge of the online education and how they are evaluated.

Another strong relation of 0.07 value it was between students needs and quality of education in our case online education, so the student involvement and virtual approach is necessary and beneficial by accessing the materials provided by teachers on platforms so the educational act not lose from his quality.

By using online platform from quality characteristics point of view the reading (R), of suplimentary and speciality materials, obtain the strogest value of 0.78.

Also for individual charactersitics of students the level of education obtain a bigest value of 0.47 thats mean that students are informed and enjoy to used the platforms and be involved in online education with the finalization on specialization.

For Needs the highest value of 0.12 was obtain by number of hours spending by students online (H), folowed by the frequency (F) with which they access the platform with a value of 0.03. For Knowledge the highest value of 0.07 was obtain by evauation using test mulitole choice and the lowest by written examination with a value of -0.9.

In conclusion between students individual characteristics and students knowledge’s using digital platforms there is a strongest correlation and the quality of educational process it is not influence by the individual characteristics. Student’s attitude towards online education it is influence by their needs and platform quality improving student’s knowledge’s and behavior.

5.3. Conventional Students Cluster

In order to be able to study students’  behavior regarding the online education, a cluster analysis was also performed this time taking into account the order of their preferences. The sample of 300 students was subjected to a k mean and hierarchical grouping, we proceeded  to identify four groups of students with similar educational profile. According to the features of the adult students (Table 10) four clusters were identified; needs of students (characterized by online benefits, and individual study of specific materials), students knowledge (characterized by the hours spend weekley online ) ,quality for   students (characterized by quality of materials and speciality materials,), students preference for evaluation (characterized by  different types of online evaluation, multiple choice, portofolio and writen version).Table 10.  Students conventional clusters

Items

Clusters

Number of Students

Gender

Needs

73

Education level

Do you consider beneficial online education

Do you study supplementary material propose by teachers

Age

Knowledge

73

How many hours are you spending weekly online

Do you read the specialty materials for a better understanding

Quality

32

What  kind of  examination  do you  prefer

Preferences

122

Cluster 1: for 73 students needs of online education and supplementary materials post on platform by teachers express their attitude to spend time and hours to colect the informations. 

Cluster 2: for 73 respondents knowledge’s are important, the number of hours spent on online or on platforms depend their information’s and enter in connection with teachers and participate on different activities. Age confirms once again student’s responsibilities for their professional preparation and skills impact in using new technology.

Cluster 3: quality obtain the lowest value for using additional materials. The low value shows that teachers provide enough information through online courses and materials posted on platforms. The university's own platform and for each specialization allows students access at any time to have access to information.

Cluster 4: the highest value for 122 students, was obtained in terms of cluster for the student preferences for final evaluation, during examination or periodical evaluation. Among the verification options proposed: online, multiple choice, test and written verification students have shown that assessment is very important for them in pandemic time also.

  1. Conclusion

Point 2. It is needed to clarify the objective of the study.

As per the article, the purpose of the study is to know the opinions of the students about digital learning in their daily life (217) and/or their opinion about digital learning during the pandemic period (228).

Response 2: I made  the correction.

Point 3: However, in the materials and method section, it is only specified that it is intended to know the student's appreciation during the pandemic period while the course was suspended (229), and it is not specified the digital learning in their daily life. Then, these are two different objectives that must be clarified.

Response 2:

I clarify in text

It must be determined the teaching-learning context for better understanding how the students were affected by the COVID pandemic situation since, as the article indicates, it was under this situation when the study was carried out. For this, answers to the following questions would be needed but, however, they are not clarified in the article:

Point 3. How were online classes handled during the pandemic period?  Was any online platform used at the university?

Response 3: The pandemic did not find the university and teachers totally unprepared because a platform (edu.utcluj.ro) was created long before for the students from the low frequency system, where the courses, seminar materials and other information necessary for those who work and attend the faculty were posted. The platform was also for full-time students to be able to send homework proposed by teachers or for teacher-student information. During the pandemic, the platform was improved and another one Knowledge Base (kb.cunbm.utcluj.ro) platform was created, respectively for the current ones, allowing the students from the economic specialization to access them and to build the teacher-student bridge.

In addition to the digital platform for seminars, homework, additional materials, systematic or periodic evaluation, some teachers have chosen other teaching tools, respectively ZOOM or Microsoft Team, for a more attentive and beneficial communication, in this way digital technology has brought a plus the educational act.

Point 4. Were they synchronous or asynchronous?

 Response 4. The transition from the traditional classroom teaching system but keeping the schedule, to online education was made gradually at the beginning only the courses were held online and the face-to-face seminars, but with the lockdown during the pandemic, everything moved to the total online system. The academic staff put in practice Hrastinski [81] and Flora Amiti [82] suggestion by keeping the teaching systems but using actual modern digital tools, all three modes of online learning, asynchronous, synchronous and hybrid. 

Point 5. Was the evaluation of the subjects in the university/faculty modified?

Response5. Taking the study results the evaluation remain online

Point 6.In the article, it is mentioned that to achieve this objective two processes have been carried out: interviews and a questionnaire.

Response 6. The questionnaire was applied between October- November 2020 when the pandemic restriction allowed students’ to participate in specific situations in activities at the university. The sample of 300 respondents consisted of graduate students in the final grades and master students from department of economics because they could compare the two methods of education before and during the pandemic.

The items were establish to determine the factors who are influencing the students behavior and attitude upon online education. The study wanted to identify students' behavior and their attitudes in the new context of online education, the structure of questionnaire and factors influencing their behavior are presented in Table 4 .

Point 7. In relation to the interviews carried out during the period when the face-to-face classes were on hold:

Response 7. In order to obtain the necessary data, the target group was selected on a voluntary basis and the period was targeted respectively when students chose their place, the period of the internship or the coordinators of the license or dissertation. All students voluntarily consented and confirmed their participation in the study after being previously informed of the purpose of the study.

Point 8. What was the purpose?

Response 8. The study allowed the analysis of the behavior and association with student’s attitude towards online education.

Point 9.  What questions were asked?

Response 9

Questions

Items

Factor

1

Age

I1

Individual   Characteristic

2

Gender

I2

3

Education level 

I3

4

How many hours are you spending weekly online?

H

Needs

5

Do you find digital learning and platforms useful?

U

6

How many hours do you devote to individual study?

S

14

How often you enter online

F

8

What kind of examination  do you prefer (online)

E1

Knowledge

9

What kind of examination do you prefer (writing)

E2

10

What kind of examination do you prefer (test )

E3

11

What kind of examination do you prefer synthesis

E4

12

What kind of examination  o you prefer portfolio

E5

7

Do you read the specialty materials?

R

Quality

13

How you appreciate the online courses quality

Q

15

How do you consider the learning activity

QE1

Point 10. Was there a record of the most important data carried out?

Response 10. No

Point 11. How many subjects were interviewed?

Response 11. The sample of 300 respondents consisted of graduate students in the final grades and master students from department of economics because they could compare the two methods of education before and during the pandemic

Point 12, What was the way to qualitatively analyze the data? All of these items should be mentioned in the article to better understand the value of these interviews.

Response 12.. No interview

Point 13. In relation to the questionnaire, it would be important to mention in the article the guidelines given to the students about data entry and if they were informed that this questionnaire was in the context of a research. Also, the questions are all written in the present, not in the past, then if the questionnaire was completed by the students after the classes were on hold (October-November 2020), how the author can guarantee that their responses are in relation to the pandemic period and not to their current daily life.

Response 13. In order to obtain the necessary data, the target group was selected on a voluntary basis and the period was targeted respectively when students chose their place, the period of the internship or the coordinators of the license or dissertation. All students voluntarily consented and confirmed their participation in the study after being previously informed of the purpose of the study.

Point 14. The structure of the questionnaire is not properly explained and the variables evaluated are confusing. For example:

  • In Table 2, which shows the structure of the questionnaire, Students Knowledge items are discussed in the second part.
  • However, in the explanation of the table content, part 2 refers to Student's Needs (243), which is in fact the part three in the table, and even later (270) it is referring to Student Behaviour, which is part four in the table.

Response 14/ I amke the changes to be same in enter article.

Questions

Items

Factor

1

Age

I1

Individual   Characteristic

2

Gender

I2

3

Education level 

I3

4

How many hours are you spending weekly online?

H

Needs

5

Do you find digital learning and platforms useful?

U

6

How many hours do you devote to individual study?

S

14

How often you enter online

F

8

What kind of examination  do you prefer (online)

E1

Knowledge

9

What kind of examination do you prefer (writing)

E2

10

What kind of examination do you prefer (test )

E3

11

What kind of examination do you prefer synthesis

E4

12

What kind of examination  o you prefer portfolio

E5

7

Do you read the specialty materials?

R

Quality

13

How you appreciate the online courses quality

Q

15

How do you consider the learning activity

QE1

  • In the explanation (page 7, 288), it is explained that part 4 refers to the preferences in the evaluation, while in the table it refers to the Students Perception.
  • In the development of the explanation (270) it is specified that the attitudes of the students are established (it is not named in table 2) what questions exactly evaluate the attitude?

Response 14 The study present the following four factors that influence students behavior were taken into consideration:

  1. Student’s individual characteristics (age, gender, education level);
  2. Student’s needs: frequency of entering on online platform (F), hours spending on virtual platform (H), hours to study and learn from materials propouse by teachers (S), and how usuful are digital platforms for their needs and for their better understanding (U);
  3. Student’s knowledges regarding the type of knowledges evaluation portofolio, syntesis, test, written examination and online examiination (E1, E2, E3, E4 and E5).
  4. Student’s perception about quality of online education: courses quality (Q), education and learning quality (QE1) and quality of materials presented and informations (R).

The research model from Figure 2 is based on the research objectives and hypothesis.

Figure 1. Research model to examine the students behavior and attitude

Point 15. Following the explanation of the questionnaire (288), the author comments again that there are three items in relation to the evaluation, but in the structure of Table 2 the three final questions do not refer to the evaluation and are related to the student's perception in relation to the courses.

Response 15. I made the corrections.

Point 16. In conclusion, when reading the design, there are doubts in relation to the structure of the questionnaire and whether the questions asked really evaluate what it intends to measure.

Response 18. Corrcetions done

Point 17. It is recommended to clearly define each variable. Validation by an expert would have been necessary to assess the questionnaire and to make sure that the items evaluate what is really intended in relation to online education.

Response 17. Done

Point 18. Also, it is necessary to explain the differences between each variable and with what specific items each is evaluated.

Response 18. Done

Questions

Items

Factor

1

Age

I1

Individual   Characteristic

2

Gender

I2

3

Education level 

I3

4

How many hours are you spending weekly online?

H

Needs

5

Do you find digital learning and platforms useful?

U

6

How many hours do you devote to individual study?

S

14

How often you enter online

F

8

What kind of examination  do you prefer (online)

E1

Knowledge

9

What kind of examination do you prefer (writing)

E2

10

What kind of examination do you prefer (test )

E3

11

What kind of examination do you prefer synthesis

E4

12

What kind of examination  o you prefer portfolio

E5

7

Do you read the specialty materials?

R

Quality

13

How you appreciate the online courses quality

Q

15

How do you consider the learning activity

QE1

Point 19. -Reference number 11 is cited with different authors in the text (79) (213) see (669).

Response 19.  Change and made the correction in text

Reviewer 2 Report

The paper is very interesting and, of course, very topical. Congratulations. However, I believe there are some considerations that could help improve it:

Abstract. Lines 8-9. This idea is more appropriate for a conclusion than an introduction.

Introduction. Figure 1. Consider delete it. Information presented in lines 55-87 and table 1 are clear enough.

The way in which references are presented in the introduction section should be reviewed, according to the journal guidelines (reference numbers should be placed in square brackets [ ]). 

Line 201. Delete it. The objectives must be presented at the end of the introduction section without a separate section.

Materials and methods. Informed consent has been obtained? Please explain in lines 222-226 how students were involved into the study. Was this done voluntarily?

Lines 341-342. It may be interesting to add a reference to reinforce the comment on the Cronbach's alpha value obtained.

Line 440. Check word spacing.

Lines 532-643. In accordance with the guidelines for the author, a discussion section and a conclusion section are necessary. Therefore, the information shown in the latter parts of the paper should be restructured.

Author Response

Response to Reviewer 2 Comments

Dear Editor

Dear Reviewer

I very much appreciated the encouraging, critical and constructive comments on this manuscript by the reviewer. The comments have been very thorough and useful in improving the manuscript. I strongly believe that the comments and suggestions have increased the scientific value of revised manuscript by many folds. I have taken them fully into account in revision. I am submitting the corrected manuscript with the suggestion incorporated the manuscript. The manuscript has been revised as per the comments given by the reviewer, and our responses to all the comments are as follows:

R2

The paper is very interesting and, of course, very topical. Congratulations. However, I believe there are some considerations that could help improve it:

Point 1:  Abstract. Lines 8-9. This idea is more appropriate for a conclusion than an introduction.

Response 1: I make the correction  

Abstract: Universities around the world have faced a new pandemic that has forced closure unlimited term campuses and conducting educational activities on online platforms. The paper presents a survey about students behavior and attitude towards online education in pandemic period from Technical University of Cluj Napoca, Romania. A group of 300 students participated. The questionnaire was structured in four parts to determine student’s individual characteristics, student’s needs, students’ knowledge’s in using virtual platforms and students quality preferences for education online. Students said that online education in a pandemic situation is beneficial for 78% of students. 41.7% percent from students appreciate teachers' teaching skills and the quality of online courses from the beginning of the pandemic. Students appreciated in 18.7% percent the additional materials online for study to support education. However, students found online education stressful but preferred online assessment for evaluation. This pandemic has led to the new stage of Education 4.0 online education and the need to harmonize methods of education with the requirements of new generations

Point 2:  Introduction. Figure 1. Consider delete it. Information presented in lines 55-87 and table 1 are clear enough

Response 2: I delete the Figure 1

Point 3. The way in which references are presented in the introduction section should be reviewed, according to the journal guidelines (reference numbers should be placed in square brackets [ ]). 

Response 3: I made the correction.

Point. Line 201. Delete it. The objectives must be presented at the end of the introduction section without a separate section.

Response 3: I made the correction

Point 4. Materials and methods. Informed consent has been obtained?

Response 4: I made the correction

Point 5.  Please explain in lines 222-226 how students were involved into the study. Was this done voluntarily? 

Response 1: The questionnaire was applied between October- November 2020 when the pandemic restriction allowed students’ to participate in specific situations in activities at the university. The sample of 300 respondents consisted of graduate students in the final grades and master students from department of economics because they could compare the two methods of education before and during the pandemic.

The items were establish to determine the factors who are influencing the students behavior and attitude upon online education. The study wanted to identify students' behavior and their attitudes in the new context of online education, the structure of questionnaire and factors influencing their behavior are presented in Table 4 .

In order to obtain the necessary data, the target group was selected on a voluntary basis and the period was targeted respectively when students chose their place, the period of the internship or the coordinators of the license or dissertation. All students voluntarily consented and confirmed their participation in the study after being previously informed of the purpose of the study. The study allowed the analysis of the behavior and association with student’s attitude towards online education

Point 6. Lines 341-342.

It may be interesting to add a reference to reinforce the comment on the Cronbach's alpha value obtained.

Response 6. Taking in consideration Saraçli et al.[83] and Gumus et al.[84] papers the Cronbach’s alpha were calculated as a quality instrument to analyze the items. The total Cronbach’s alpha value of scale was calculated as 0.72 which is statistically one of the indicators that shows that the reliability of the scale. With α value in interval 0.8 ≤ α ≥ 0.7 interval  following the statistical rule [85],[86] we can consider acceptable and good enough the model. After reliability analysis, exploratory factor analysis (EFA) was applied over the for four factors Individual charactersitic of students, Needs, Knowledges and Quality.

 Using the Schermelleh-Engel and Moosbrugger structural model [87] the next step was to compare with the program solution, For structural equation model we obtain the values from Table 5. The RMEA: root mean square error of approximation value was equal with 0.63, NFI: normed fit index equal with 0.93, NNFI: non-normed fit index equal with 0.96, CFI: comparative fit index egual with 0.90, GFI: goodness of fit index egual with 0.90 and AGFI: adjusted goodness of fit index was equal with 0.88. If we compare with the goodness fit the model is in acceptable fitness.

Table 5.  Structural mode fit

Criteria

Acceptable Fitness

Model

RMSEA

0.05 ≤ RMSEA ≤ 0.10

0.63

NFI

0.90 ≤ NFI ≤ 0.95

0.93

NNFI

0.95 ≤ NNFI ≤ 0.97

0.96

CFI

0.95 ≤ CFI ≤ 0.97

0.90

GFI

0.90 ≤ GFI ≤ 0.95

0.90

AGFI

0.85 ≤ AGFI ≤ 0.90

0.88

Source: Adaptation after Schermelleh-Engel and Moosbrugger [87]

Point 7:  Line 440. Check word spacing.

Response 7: I done the correction

Point 8:  Lines 532-643. In accordance with the guidelines for the author, a discussion section and a conclusion section are necessary. Therefore, the information shown in the latter parts of the paper should be restructured.

Response 8. I create separate Section   5. Dissscusion and  6. Conclusion

  1. Discussion

5.1. Variables Correlation

To determine the correlation between the variables and to obtain the degree to which online education influences students’ behavior, the classification and regression tree (CRT) analysis was used. Figure 2 shows the classification of students’ behavior regarding the information they obtain using supplementary materials for a better understanding of the information received but also the benefits of online education.

Figure 2. Classification for students behavior using supplimentary materials. Source: By author.

Figure 3, present the classification for students’ behavior using education online. Weekly or daily is a routine for 24.7 % of respondents to spend more than 10 hours on platform and 58.7% of students are spending between 1-10 hours daily.

Figure 3. Classification for student’s behavior using online education.

Additionally, 26% of students are male and 32.7 % are female. Taking into account the time spent by students online on educational platforms, we notice that a percentage of 57.8% of students spend between 1-10 hours per week, of which 43.2% are at the bachelor's level and 56.8% are at the master's level, we note that they use the information on the platforms put on display by the teachers.

The highest value was obtained for undergraduate students who spend more than 10 hours on the university platform, respectively for the field of management and economics in a percentage of 24.4%.

5.2. Correlations between Items

Using the database obtain from the 15 items of questionnaries from SPSS using the Lisrel 8.7 program, four groups of students with similar profiles regarding the online education were identify.

The model take into consideration the students characteristics (gender, age, education level), how the students are involved in diferent activities for a better asimilation of informations and their knowledges from different activities for their culture (U, S, H, F), students needs for evaluation (E1,E2,E3,E4 and E5) and students quality satisfaction about online education (R, Q, QE1). The program solution for the students behavior model it is presented in Figure 4.

Figure 4. Factors influencing students’ behavior using online education

           Students behavior using online education towards pandemic period gives us the following correlation:

The highest positive value of 0.89 was obtained for the correlation between individual characteristics and students knowledge. The results present that students from Baia Mare faculty, the future managers and economists or entrepreneurs, have strong knowledge of the online education and how they are evaluated.

Another strong relation of 0.07 value it was between students needs and quality of education in our case online education, so the student involvement and virtual approach is necessary and beneficial by accessing the materials provided by teachers on platforms so the educational act not lose from his quality.

By using online platform from quality characteristics point of view the reading (R), of suplimentary and speciality materials, obtain the strongest value of 0.78.

Also for individual charactersitics of students the level of education obtain a bigest value of 0.47 thats mean that students are informed and enjoy to used the platforms and be involved in online education with the finalization on specialization.

For Needs the highest value of 0.12 was obtain by number of hours spending by students online (H), folowed by the frequency (F) with which they access the platform with a value of 0.03. For Knowledge the highest value of 0.07 was obtain by evauation using test mulitole choice and the lowest by written examination with a value of -0.9.

In conclusion between students individual characteristics and students knowledge’s using digital platforms there is a strongest correlation and the quality of educational process it is not influence by the individual characteristics. Student’s attitude towards online education it is influence by their needs and platform quality improving student’s knowledge’s and behavior.

5.3. Conventional Students Cluster

In order to be able to study students’  behavior regarding the online education, a cluster analysis was also performed this time taking into account the order of their preferences. The sample of 300 students was subjected to a k mean and hierarchical grouping, we proceeded  to identify four groups of students with similar educational profile. According to the features of the adult students (Table 10) four clusters were identified; needs of students (characterized by online benefits, and individual study of specific materials), students knowledge (characterized by the hours spend weekley online ) ,quality for   students (characterized by quality of materials and speciality materials,), students preference for evaluation (characterized by  different types of online evaluation, multiple choice, portofolio and writen version).Table 10.  Students conventional clusters

Items

Clusters

Number of Students

Gender

Needs

73

Education level

Do you consider beneficial online education

Do you study supplementary material propose by teachers

Age

Knowledge

73

How many hours are you spending weekly online

Do you read the specialty materials for a better understanding

Quality

32

What  kind of  examination  do you  prefer

Preferences

122

Cluster 1: for 73 students needs of online education and supplementary materials post on platform by teachers express their attitude to spend time and hours to colect the informations. 

Cluster 2: for 73 respondents knowledge’s are important, the number of hours spent on online or on platforms depend their information’s and enter in connection with teachers and participate on different activities. Age confirms once again student’s responsibilities for their professional preparation and skills impact in using new technology.

Cluster 3: quality obtain the lowest value for using additional materials. The low value shows that teachers provide enough information through online courses and materials posted on platforms. The university's own platform and for each specialization allows students access at any time to have access to information.

Cluster 4: the highest value for 122 students, was obtained in terms of cluster for the student preferences for final evaluation, during examination or periodical evaluation. Among the verification options proposed: online, multiple choice, test and written verification students have shown that assessment is very important for them in pandemic time also.

  1. Conclusion

Reviewer 3 Report

Review of sustainability-1246121

The manuscript, factors influencing student’s behavior and attitude towards online education during COVID-19, presents a study conducted in a university in Rumania regarding students’ attitudes toward online education. The author proposes to call Education 4.0 to the present education wave of the use of online education due to COVID-19 restrictions. This is, indeed, a novel idea. However, there is no research that backs up that assumption nor does the author explain what the previous three education waves were. In terms of review of existing research, the author presents ideas from many different sources, although it is difficult to understand what the author is trying to convey. Some examples are on lines 70-71, “The influence of the pandemic after Cao et al. [6], Carroll and Conboy [7], led to new practices that emerged under the pressure of the pandemic” but the new practices are not discussed. Also on lines 75-78, “Also the inequalities were identifying in their research by Beaunoyer et al. [8], Becker et al. [9] where electronic platforms allow the storage and management of an unlimited number of courses, as well as the storage and management of an unlimited volume of content within a course.” What were the identified inequalities? What does the ability to store courses online inform such inequalities? There are multiple instances of this lack of explanation, before, in, and after the Review of Literature section, that prevents the reader from fully understanding what the author is trying to convey.

In terms of research design, question, hypotheses, and methods, this reviewer has some observations/questions that stem from a lack of clarity in the manuscript. The first and most obvious question is, why does the research design use surveys applied in a face-to-face format during the pandemic? If the author was trying to research the online learning environment, why doing the research in a face-to-face format? The questionnaire used is divided into four parts presented in Table 2 (line 238) but the explanation of each part, presented below the table, does not mention the parts by the same name as they are in the table (lines 246-250).  There are five hypotheses presented in the study. However, H4 seems to be stated in an inverse way to the rest of the hypotheses (l. 307-308). Next, the author presents a nice research model that she seemed to have validated and discussed later in the manuscript. As with previous instances, the model needs a better explanation. An example is on l. 328-329 where it stated, “The model shows the direct and indirect relationships between the research variables (Needs, Knowledge, and Evaluation).” However, the word “Evaluation” is not part of the model, at least there are no boxes with that name in the model.

In the Results section, the author included Tables 5 and 6, which have different names but have exactly the same content (lines 382 and 430). The author made a good effort in presenting results in different formats, but the explanations of such results are confusing or do not correspond to what the reader sees on such tables or figures. One figure, in particular, Figure 5, may be more understandable in table format if a good explanation accompanies the table. The discussion of Findings and Conclusion presents the same clarity problems as the previous sections. In fact, there are some statements that are not clearly connected to the results. Some examples are: “In conclusion between the individual characteristics and doing courses on virtual platforms there is no correlation, also the quality of educational process are not influence by the individual characteristics” (l. 492-494) but it is well known that individual characteristics influence the quality of educational processes that the learner chooses to engage in. When discussing the percentage of students who seek additional reading material for a course, the author states on lines 525-526: “The low value shows that teachers provide enough information through online courses and materials posted on platforms.” However, it could be that the students do not take the initiative to seek extra materials for the class. In other words, the author jumps to conclusions that do not obviously come from the data presented. Other instances of this disconnection can be found on lines 552-553 (“In our study the complicated situations and inequalities was confirmed” but no data or research questions are linked to the statement), and lines 580-587 where a series of statements come out of nowhere, although the author says that “As a final conclusion based on the data obtained and based on statistical observations, the following can be highlighted.” Finally, the author presents a long list of References, but it is unclear how they support the results based on the data presented.

Author Response

Response to Reviewer 3 Comments

Dear Editor

Dear Reviewer

I very much appreciated the encouraging, critical and constructive comments on this manuscript by the reviewer. The comments have been very thorough and useful in improving the manuscript. I strongly believe that the comments and suggestions have increased the scientific value of revised manuscript by many folds. I have taken them fully into account in revision. I am submitting the corrected manuscript with the suggestion incorporated the manuscript. The manuscript has been revised as per the comments given by the reviewer, and our responses to all the comments are as follows:

Review of sustainability-1246121

The manuscript, factors influencing student’s behavior and attitude towards online education during COVID-19, presents a study conducted in a university in Rumania regarding students’ attitudes toward online education.

The author proposes to call Education 4.0 to the present education wave of the use of online education due to COVID-19 restrictions. This is, indeed, a novel idea.

Point 1: 

However, there is no research that backs up that assumption nor does the author explain what the previous three education waves were

Response 1: The concept of education has changed dramatically over the last few years with many questions being raised as to what the best mode of instruction is with the advent of technology and the Internet. The waves of the evolution of education in time begin in the 1780’s as the first wave in the individual context of learning and memorization for the experience, known as Education 1. The second wave of mass learning appears around the 1900’s known as Education 2 or the Internet that allows learning. Education 3, begin from the 1970’s and brings computers, but only as an interface with students which produces knowledge. Distance learning was first introduced in the 18th century in parallel with the postal service, but it didn’t pick up steam until communications technology evolved in the 1990’s [1]. If we look in time at the stages of the evolution of education, we can see that from a traditional system that focused on books and teaching on the blackboard, over time the use of technology induced a new stage known as Education 4.0 when the computer and the Internet changed the concept of education and the new digital generation offers more possibilities for education,

The future belongs to Education 4.0 as a part of the evolving stage but with a very high impact of digital technology. Empowering education to improve innovation the transition to the new stage requires the development and harmonization of education systems by calling the new relationship that must be established student-teacher-technology = smart education and the use of e-education (online, electronic tools). Zhu et al. [2] are supporters of smart education for an environment in which students work as close as possible to reality, which is reason the education system must combine reality with the virtual world. Zhu et al [2] and Hartono et al. [3], set out the needs for hybrid education and the term smart learning for students to adapt the education to the digital age.

Point 2:  In terms of review of existing research, the author presents ideas from many different sources, although it is difficult to understand what the author is trying to convey.

Some examples are on lines 70-71,

“The influence of the pandemic after Cao et al. [6], Carroll and Conboy [7], led to new practices that emerged under the pressure of the pandemic” but the new practices are not discussed.

Response 2:  I create new section and subsections for a better understanding 

  1. Literature Review

2.1. Online learning and education

The use of the Internet and state-of-the-art technologies to obtain information and for fast communication has become extremely important in the communication and promotion strategy of any university [27]. Communication in the university environment is one of the basic elements on which the student - teacher - university relationship is built. The motivation to approach the communication made by universities starts from the premise that most of the times the students' performances in the learning process and in the one of integration in the university environment are determined by the way in which the information is made by the universities. Moreover, the COVID-19 pandemic has demonstrated the usefulness of these platforms as more and more schools move to the red scenario, which means that virtually the entire educational process moves to the online system on educational teaching and learning platforms.

2.2. Online education and teachers

Universities and teachers were not completely taken aback by online courses and activities and Windes and Lesht [45] highlighted the effects of online courses and their impact on education.

There are currently few studies on the effectiveness of online courses, the teacher-student relationship, and the effectiveness of online assessments. Among those who approached the new topic were Pinaki et al [46] and A. Patricia [47] who noticed that students believe that online education helped them to continue their training and studies during the pandemic with digital platforms, but at the same time to have access to faculty libraries.

But online education for teachers requires time to identify and build the platforms and materials needed according to Hodges et al. [48]. Bojovic et al. [39] and Pinaki et al [46] noted that teachers still lack confidence on online assesment techniques.

2.3. Students behavior and attitude toward online education

Since 1986 when appear the first model Technology Acceptance Model (TAM) to identify the factors that affect the students behavior and intention to use technology, in time the model was improved and new factors and a more complex investigation was develop like in Table 2.

Point 3:  Also on lines 75-78,

“Also the inequalities were identifying in their research by Beaunoyer et al. [8], Becker et al. [9] where electronic platforms allow the storage and management of an unlimited number of courses, as well as the storage and management of an unlimited volume of content within a course.

” What were the identified inequalities?

Response 3: The study identify students needs, this data suggests that it is not very realistic to start from the assumption that switching to teaching exclusively online can be done easily. The study identify and confirm also that are some inequalities regarding the internet access (no telephone signal, or do not have a computer / laptop / tablet / mobile phone, as well as a fairly low level of digital skills) [8], [18].Students have begun to notice that it is possible to put in more effort and more time to attend courses and applications through online digital tools, even if they are in isolation, at home.

Point 4:   What does the ability to store courses online inform such inequalities?

There are multiple instances of this lack of explanation, before, in, and after the Review of Literature section, that prevents the reader from fully understanding what the author is trying to convey.

Response 4. I've done the modification

The study identify students needs, this data suggests that it is not very realistic to start from the assumption that switching to teaching exclusively online can be done easily. The study identify and confirm also that are some inequalities regarding the internet access (no telephone signal, or do not have a computer / laptop / tablet / mobile phone, as well as a fairly low level of digital skills) [8], [18].Students have begun to notice that it is possible to put in more effort and more time to attend courses and applications through online digital tools, even if they are in isolation, at home.

In terms of research design, question, hypotheses, and methods, this reviewer has some observations/questions that stem from a lack of clarity in the manuscript. The first and most obvious question is, why does the research design use surveys applied in a face-to-face format during the pandemic?

Point 5. If the author was trying to research the online learning environment, why doing the research in a face-to-face format?

Response 5: I've done the modification

The pandemic did not find the university and teachers totally unprepared because a platform (edu.utcluj.ro) was created long before for the students from the low frequency system, where the courses, seminar materials and other information necessary for those who work and attend the faculty were posted. The platform was also for full-time students to be able to send homework proposed by teachers or for teacher-student information. During the pandemic, the platform was improved and another one Knowledge Base (kb.cunbm.utcluj.ro) platform was created, respectively for the current ones, allowing the students from the economic specialization to access them and to build the teacher-student bridge.

In addition to the digital platform for seminars, homework, additional materials, systematic or periodic evaluation, some teachers have chosen other teaching tools, respectively ZOOM or Microsoft Team, for a more attentive and beneficial communication, in this way digital technology has brought a plus the educational act.

The transition from the traditional classroom teaching system but keeping the schedule, to online education was made gradually at the beginning only the courses were held online and the face-to-face seminars, but with the lockdown during the pandemic, everything moved to the total online system. The academic staff put in practice Hrastinski [81] and Flora Amiti [82] suggestion by keeping the teaching systems but using actual modern digital tools, all three modes of online learning, asynchronous, synchronous and hybrid. 

The questionnaire was applied between October- November 2020 when the pandemic restriction allowed students’ to participate in specific situations in activities at the university. The sample of 300 respondents consisted of graduate students in the final grades and master students from department of economics because they could compare the two methods of education before and during the pandemic.

The items were establish to determine the factors who are influencing the students behavior and attitude upon online education. The study wanted to identify students' behavior and their attitudes in the new context of online education, the structure of questionnaire and factors influencing their behavior are presented in Table 4 .

Table 4.  Questionnaire structure.

Questions

Items

Factor

1

Age

I1

Individual   Characteristic

2

Gender

I2

3

Education level 

I3

4

How many hours are you spending weekly online?

H

Needs

5

Do you find digital learning and platforms useful?

U

6

How many hours do you devote to individual study?

S

14

How often you enter online

F

8

What kind of examination  do you prefer (online)

E1

Knowledge

9

What kind of examination do you prefer (writing)

E2

10

What kind of examination do you prefer (test )

E3

11

What kind of examination do you prefer synthesis

E4

12

What kind of examination  o you prefer portfolio

E5

7

Do you read the specialty materials?

R

Quality

13

How you appreciate the online courses quality

Q

15

How do you consider the learning activity

QE1

Point 6:  The questionnaire used is divided into four parts presented in

Table 2 (line 238) but the explanation of each part, presented below the table, does not mention the parts by the same name as they are in the table (lines 246-250).

Response 6.

In order to obtain the necessary data, the target group was selected on a voluntary basis and the period was targeted respectively when students chose their place, the period of the internship or the coordinators of the license or dissertation. All students voluntarily consented and confirmed their participation in the study after being previously informed of the purpose of the study. The study allowed the analysis of the behavior and association with student’s attitude towards online education.

The first part of questionnaire tried to identify the socio-demographic characteristics of the respondents, three questions were used regarding gender, age and education level. The education level includes two categories: bachelor and master students. The age groups were establish between 18-26 years or younger and 26-32, 32–38, and 38-42 years of age.

The second part of questionnaire were focused on students behavior related to their needs: time spend using online education tools. Additionally, the questionnaire establishes and identifies the frequency of hours of  consumption of virtual tools used in education, which was measured by asking respondents: ’’How often you enter weekly online ?’’. Response categories for their time spend on the online  to read suplimentary materials (1 hour, 10 hours, more then 10 hours) and the emphasis they put on the quality of courses  and how this affects students behavior.

In the third section of the questionnaire, students knowledges were established, to determine students culture on virtual media and their abilities to use the modern tools, also the benefits of education online, and how important for students it’s education and their orientation between traditional and new type of education. The answer categories for the open question about type of examination prefered by students offered the respondents the possibility of stating their favorite one. The key element was the item regarding the final evaluation of their activity, respectively the way of conducting the final evaluation exam. They were able to choose from several types of assessment so as to see which type of assessment is considered the most optimal assessment of their knowledge. Regarding the students preferences type of examination, it was possible to choose between online, writitng , multiple choise test, sintesis and protofolio. A Likert scale was used for the study, starting from a score of 1 represented the student disagreement "Not at all ‘’ and up to a score of 5 represented very well represented the student strongly agreement.

The last part of the questionnaire included three questions that identify students   satisfaction about the quality of online courses, the quality of  learning activity on the pandemic period and the quality of specialty materials. From research the article wanted to obtain and know students interaction with the professor through an online platform  quality with the item ‘’How you appreciate the online courses quality’’, how usefulness is online education was identified with the item ‘’Do you find digital learning and teaching platforms useful?’’

Point 7:  There are five hypotheses presented in the study.

However, H4 seems to be stated in an inverse way to the rest of the hypotheses (l. 307-308).

Response 7: corrections done

The study present the following four factors that influence students behavior were taken into consideration:

  1. Student’s individual characteristics (age, gender, education level);
  2. Student’s needs: frequency of entering on online platform (F), hours spending on virtual platform (H), hours to study and learn from materials propouse by teachers (S), and how usuful are digital platforms for their needs and for their better understanding (U);
  3. Student’s knowledges regarding the type of knowledges evaluation portofolio, syntesis, test, written examination and online examiination (E1, E2, E3, E4 and E5).
  4. Student’s perception about quality of online education: courses quality (Q), education and learning quality (QE1) and quality of materials presented and informations (R).

The research model from Figure 2 is based on the research objectives and hypothesis.

Needs

Students Knowledge’s

Individual characteristics

Quality

Students

Attitude

Students

Behavior

Education Preferences

Figure 1. Research model to examine the students behavior and attitude. Source: By author

The hypotheses tested on the attitude of students in the present study are:

Hypotesis 1.(H1): Students preferences have a significant effect on their attitudes towards their online  education.

Hypotesis 2 (H2): Students preferences have a significant effect on their attitudes towards needs.

Hypotesis 3.(H3): Students preferences have a significant effect on their attitudes towards knowledge.

Hypotesis 4.(H4): Attitudes of students concerning their needs of evaluation has a significant effect on their behavior.

Hypotesis 5.(H5): Attitudes of students towards online education have a significant effect on their behavior.

Point 8. Next, the author presents a nice research model that she seemed to have validated and discussed later in the manuscript.

“The model shows the direct and indirect relationships between the research variables (Needs, Knowledge, and Evaluation).

” However, the word “Evaluation” is not part of the model, at least there are no boxes with that name in the model.

As with previous instances, the model needs a better explanation.

An example is on l. 328-

Responce 8. Correction done

329 where it stated,

Point 9. In the Results section, the author included Tables 5 and 6, which have different names but have exactly the same content (lines 382 and 430).

Responce 9.  I delete the similar part

Table 8. Correlation between respondents age and examination needs

Age

Cumulative

Percent

18-26

26-32

32-38

38-42

What kind of examination do you prefer

Portfolio

19

5

2

0

8.67

Synthesis

17

4

3

1

8.33

Test (multiple choice)

71

20

15

9

38.33

Written exam

17

1

2

3

7.67

On-line exam

65

20

10

2

32.33

Total

189

50

 32

15

95.33

Table 9. Correlation between respondents gender and kind of examination

Gender

Cumulative

Percent

Female

Male

What kind of examination do you prefer

On-line exam

51

46

32,33

Written exam

11

12

7.67

Test (multiple choice)

70

45

38,33

Synthesis

13

12

8.33

Portfolio

16

10

8.67

Total

161

125

95.33

Points 10.  The author made a good effort in presenting results in different formats, but the explanations of such results are confusing or do not correspond to what the reader sees on such tables or figures.

Responce 10. I've done the modification

Point 11. One figure, in particular, Figure 5, may be more understandable in table format if a good explanation accompanies the table.

           Students behavior using online education towards pandemic period gives us the following correlation:

The highest positive value of 0.89 was obtained for the correlation between individual characteristics and students knowledge. The results present that students from Baia Mare faculty, the future managers and economists or entrepreneurs, have strong knowledge of the online education and how they are evaluated.

Another strong relation of 0.07 value it was between students needs and quality of education in our case online education, so the student involvement and virtual approach is necessary and beneficial by accessing the materials provided by teachers on platforms so the educational act not lose from his quality.

By using online platform from quality characteristics point of view the reading (R), of suplimentary and speciality materials, obtain the strogest value of 0.78.

Also for individual charactersitics of students the level of education obtain a bigest value of 0.47 thats mean that students are informed and enjoy to used the platforms and be involved in online education with the finalization on specialization.

For Needs the highest value of 0.12 was obtain by number of hours spending by students online (H), folowed by the frequency (F) with which they access the platform with a value of 0.03. For Knowledge the highest value of 0.07 was obtain by evauation using test mulitole choice and the lowest by written examination with a value of -0.9.

In conclusion between students individual characteristics and students knowledge’s using digital platforms there is a strongest correlation and the quality of educational process it is not influence by the individual characteristics. Student’s attitude towards online education it is influence by their needs and platform quality improving student’s knowledge’s and behavior.

Point 12. The discussion of Findings and Conclusion presents the same clarity problems as the previous sections. In fact, there are some statements that are not clearly connected to the results.

Some examples are:

“In conclusion between the individual characteristics and doing courses on virtual platforms there is no correlation, also the quality of educational process are not influence by the individual characteristics” (l. 492-494) but it is well known that individual characteristics influence the quality of educational processes that the learner chooses to engage in.

Response 12.

  1. 6. Conclusion

The article presents the results obtained following the application of questionnaires apply to identify student’s behavior and attitude towards online education during the COVID-19 pandemic.

Based on the literature the results were able to create a student profile model and establish the factors which influence student’s behavior and attitude upon online education. Online education has been a great challenge for both teachers and students. At present, education is still in a period of adaptation, of identifying the factors that influence the educational act for an as yet unexplained period. The COVID-19 pandemic brought for the first time the widespread adoption of online education around the world, making it a necessity in difficult times.

The four factors, taking in consideration for the model were the individual characteristics specific to each student, the students' knowledge, the students' needs and the preferences for the quality of online education influence the students' behavior and attitude. The student’s behavior is influence by their attitude regarding their needs and quality digital education. The students preferences for quality platforms and materials in the changes of this period, confirm the hypothesis and the model. Students’ present that teachers are those who adapt and reformulate their habits, being closer to students, through the digital environments of the future, even if further as physical distance, which contributes to a categorical evolution of university education?

The feedback of questionnaire confirms that it is strong connection between student’s needs and quality education and teaching process not only in pandemic period which influence the student’s behavior [3], [4], [11].

The results confirm students’ knowledge’s obtain by using online education during this pandemic as a useful lesson during future demands which confirm the other researcher’s results [46], [51], [52].

The study identify students needs, this data suggests that it is not very realistic to start from the assumption that switching to teaching exclusively online can be done easily. The study identify and confirm also that are some inequalities regarding the internet access (no telephone signal, or do not have a computer / laptop / tablet / mobile phone, as well as a fairly low level of digital skills) [8], [18].Students have begun to notice that it is possible to put in more effort and more time to attend courses and applications through online digital tools, even if they are in isolation, at home.

Also additionally, individual characteristics present similar preferences regarding the digital education and needs the correlation between individual characteristics and needs was confirmed [38], [40], [42].

The data obtained on the basis of statistical observations can highlight the attitude towards the use of a mixture of educational tools to ensure a new orientation towards a new vision for the future of Education 4.0 that changes their behavior towards online education not only in crises or pandemics (Table 11).

Point 13. When discussing the percentage of students who seek additional reading material for a course, the author states on lines 525-526:

“The low value shows that teachers provide enough information through online courses and materials posted on platforms.”

Response 13. I've done the modification

Point 14. However, it could be that the students do not take the initiative to seek extra materials for the class. In other words, the author jumps to conclusions that do not obviously come from the data presented.

Other instances of this disconnection can be found on

lines 552-553

(“In our study the complicated situations and inequalities was confirmed” but no data or research questions are linked to the statement), and lines 580-587 where a series of statements come out of nowhere, although the author says that “As a final conclusion based on the data obtained and based on statistical observations, the following can be highlighted.”

Response 14. I've done the modification

Point 15. Finally, the author presents a long list of References, but it is unclear how they support the results based on the data presented.

Responce 15. I've done the modification

Round 2

Reviewer 1 Report

The article has been modified considerably.
- The purpose of the study has been clarified
- The context has been clarified
- The theoretical framework has been modified in relation to the objective of the study.
- The variables of the questionnaire and the hypotheses have been clarified
- Conclusions section has been added and has been developed in accordance with the objective of the study.
- Article is now consistent.

Reviewer 3 Report

Review of sustainability-1246121_v2

The article, “Factors influencing student’s behavior and attitude towards online education during COVID-19” was substantially and positively revised. The manuscript flows better and is easier to understand. This reviewer has some clarification requests.

  1. Lines 27-28: “The second wave of mass learning appears around the 1900’s known as Education 2 or the Internet that allows learning.” Education 2 did not have an Internet that allowed learning. Maybe the author can better state that point.
  2. Lines 439-440: “Table 7, present that a 58. 66% percent from respondents spend weekly between 1-10 hours and a percent of 24.7 % more than 20 hours.” The table notes 1-10 hrs. and 10-20 hrs. but not “more than 20 hrs.”
  3. Lines 595-597: “Based on the literature the results were able to create a student profile model and establish the factors which influence student’s behavior and attitude upon online education” – it will be very useful if the author clearly states that profile before she goes on to explain it. While the explanations are good, the full profile is lost in them.

Finally, because this manuscript has so much information, the results or conclusion would be more concrete if each hypothesis is stated, whether it was accepted or rejected, and then the explanation already provided by the author.